# Short-Term Co-Application of Organic and Chemical Fertilizer Benefits Topsoil Properties and Maize Productivity in a Medium-Productivity Meadow-Cinnamon Soil

Lichao Zhai [1], Mengjing Zheng [1], Lihua Zhang [1], Jing Chen [2], Jingting Zhang [1] and Xiuling Jia [1,*]

[1] Institute of Cereal and Oil Crops, Hebei Academy of Agriculture and Forestry Science, Key Laboratory of Crop Cultivation Physiology and Green Production of Hebei Province, Shijiazhuang 050035, China; zhailichao@163.com (L.Z.); zhengmj96@126.com (M.Z.); lnzlh@126.com (L.Z.)

[2] Institute of Cotton Research, Chinese Academy of Agricultural Sciences, State Key Laboratory of Cotton Biology, Anyang 455000, China

[*] Correspondence: jiaxiuling2013@163.com; Tel.: +86-311-87670620; Fax: +86-311-87670652

**Abstract:** Co-application of organic-chemical fertilizer (CAOFCF) has attracted wide attention in China in recent years. However, its short-term effect on topsoil quality and maize yield in a medium-productivity meadow-cinnamon soil is not clear. In order to address this problem, a 3-year (2019–2021) field trial was established by arranging the following five treatments: (1) CF, applying chemical fertilizer alone; (2) OFCF1, 15% organic fertilizer + 85% chemical fertilizer; (3) OFCF2, 30% organic fertilizer + 70% chemical fertilizer; (4) OFCF3, 45% organic fertilizer + 55% chemical fertilizer; (5) OFCF4, 60% organic fertilizer + 40% chemical fertilizer. The results showed that short-term CAOFCF treatments were beneficial to the topsoil aggregate stability by increasing the percentage and mean weight diameter of macro-aggregate in topsoil. In addition, lower soil bulk density and higher soil organic carbon sequestration in topsoil were observed under the CAOFCF treatments. There was no difference in rhizosphere microbial diversity among all treatments. Compared to CF, OFCF1 and OFCF2 improved the activities of some key enzymes, including sucrase, urease, and acid phosphatase. Moreover, higher relative abundance of *Actinobacteria* and *Chloroflexi* were observed under the CAOFCF treatments. The root-shoot dry matter and maize grain yield were obviously higher in OFCF1 and OFCF2 than in CF; however, no significant difference was found in the OFCF3 and OFCF4 treatments compared to CF. The analysis of correlation suggested that there were no direct correlations between maize yield and various soil indexes measured. Nevertheless, root dry weight and root-shoot ratio were positively correlated with the activities of urease and sucrase. Meanwhile, the relationships between root dry weight, root-shoot ratio, shoot dry weight, and grain yield were all significant. In conclusion, short-term co-application of organic and chemical fertilizer (i.e., replacing 15–30% chemical fertilizer with organic fertilizer with an equal N rate) was beneficial to soil properties and maize grain yield in a medium-productivity meadow-cinnamon soil. The higher grain yield was associated with a strong maize root system, which was driven by the improved rhizosphere urease and sucrase activities.

**Keywords:** soil properties; grain yield; summer maize; organic fertilizer; chemical fertilizer





## 1. Introduction

Meadow-cinnamon soil is a typical medium-productivity soil in the northern North China Plain, which has an area of more than 0.51 million hectare. The current limited crop productivity in this area is associated with the poor soil quality resulting from long-term excessive chemical fertilizer application, including low soil organic matter, soil compaction, loss of biodiversity, and so on [1], which is not conducive to crop productivity and agricultural sustainable development.

Given that organic fertilizers are superior to chemical fertilizers for improving soil properties [2,3], more attention has focused on organic fertilizer application in field crop production [4–6]. Because the nutrient release rate in organic fertilizer is slower and the release period is also long, applying more organic fertilizer without adequate chemical fertilizer will make it difficult to meet plant requirements [7]. In this context, co-application of organic fertilizer and chemical fertilizer (CAOFCF) has been recommended as a viable solution to achieve collaborative improvement in crop productivity and soil quality.

During the past few years, CAOFCF has been confirmed as an effective measure in regulating soil quality in different agroecosystems [4,8,9], and previous research suggested that this fertilization practice can effectively improve soil physicochemical and biological properties, which is beneficial to eliminating the adverse impact caused by inadequate application of chemical fertilizer [10–12]. For example, some studies suggested that CAOFCF can regulate topsoil physical characteristics by reducing soil bulk density and improving macro-aggregate distribution [13,14], thereby promoting soil water storage capacity [15]. Moreover, soil chemical properties are also affected by CAOFCF, such as soil organic carbon (SOC) [4,13] and total N and phosphorus (P) content [11]. Aside from soil physical and chemical properties, CAOFCF also improved soil biological diversity and various soil enzyme activities [5,10–12]. In short, increasing evidence has confirmed that CAOFCF benefits soil quality and crop productivity. However, most previous studies on the positive effect of CAOFCF treatment were based on long-term field trials, whereas the short-term impact of CAOFCF on topsoil quality and maize yield in a medium-productivity meadow-cinnamon soil is rarely reported.

With organic fertilizer application gaining more attention in medium-productivity cinnamon soil area in recent years, local smallholder farmers are more concerned about the short-term effects of CAOFCF on crop productivity and sustainable farm production. Therefore, in order to clarify the short-term effects of CAOFCF on topsoil quality and farmland productivity, a short-term (3 years) field positioning trial was established with the following objectives: (1) determine the short-term effects of CAOFCF on the physicochemical properties, enzyme activities, and biodiversity of topsoil; (2) clarify the short-term effect of CAOFCF on summer maize productivity; (3) explore whether there is a relationship between soil property indicators and maize productivity as influenced by short-term CAOFCF treatment.

## 2. Materials and Methods

### 2.1. Experimental Site

The field trial was set up in 2019 at Dishang experimental station of Hebei Academy of Agricultural and Forestry Science in the northern North China Plain (37°95′ N, 114°71′ E), Gaocheng County, Hebei Province, China (Figure 1). The climate at the test site is characterized by a warm, temperate, subhumid climate, the annual mean precipitation is approximately 484 mm, and the annual mean temperature is 12.8 °C. The basic nutrient characteristics of the topsoil layer (0–20 cm) before the 2019 growing season comprised 1.87% organic matter, 1.04 g kg$^{-1}$ total N, 80 mg kg$^{-1}$ available N, 21.4 mg kg$^{-1}$ available P, and 113.9 mg kg$^{-1}$ available K, respectively.

### 2.2. Field Experimental Design

Five treatments were set up in this experiment, and the trial was arranged in a randomized block design with three replicates. The five treatments were annually applied: (1) CF, applying chemical fertilizer alone; (2) OFCF1, co-application of 15% organic fertilizer and 85% chemical fertilizer; (3) OFCF2, co-application of 30% organic fertilizer and 70% chemical fertilizer; (4) OFCF3, co-application of 45% organic fertilizer and 55% chemical fertilizer; (5) OFCF4, co-application of 60% organic fertilizer and 40% chemical fertilizer. All treatments in this study actually replaced the partial chemical fertilizer with organic fertilizer with an equal N rate (i.e., the total N, P$_2$O$_5$, and K$_2$O provided by the organic and chemical fertilizers of each treatment remained constant), and all treatments included

225 kg ha$^{-1}$ N, 120 kg ha$^{-1}$ P$_2$O$_5$, and 120 kg ha$^{-1}$ K$_2$O. The area of the experimental plot was 130 m$^2$. Maize hybrid YD9953 was used as material in this study, and the plant density was 75,000 plants ha$^{-1}$. In the 2019, 2020, and 2021 growing seasons, 60% N, total P$_2$O$_5$, and K$_2$O were applied as basal fertilizer for each treatment, and 40% N was topdressed at the V8 (eight-leaf) stage. In this study, the organic fertilizer in all treatments was applied as basal fertilizer, and the basal organic and chemical fertilizers were incorporated into the soil at a depth of 20 cm using a rotary tiller just before sowing. Other field management procedures were the same as those applied by the local smallholder farmers.

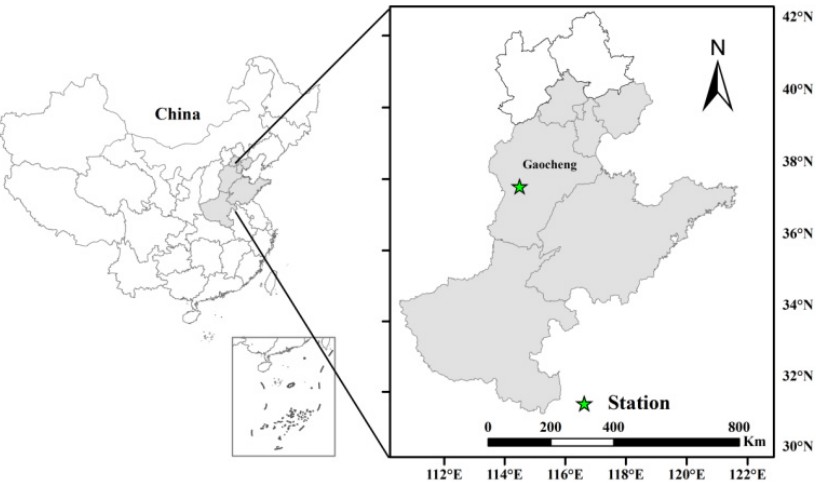

**Figure 1.** The location of experimental site. The left shows the location of the North China Plain while the right shows the location of the experimental site (green start) within the North China Plain.

### 2.3. Soil Sampling and Measurement

#### 2.3.1. Soil Sampling

Soil samples were collected from the 0–10 and 10–20 cm soil layers at harvest in 2021 to analyze the macro-aggregates, soil porosity, and SOC content. The samples for macro-aggregates and SOC were collected using a 5 cm diameter auger; the samples for soil bulk density and porosity were collected using a steel cylinder of 100 cm$^3$ volume. The rhizosphere soil was sampled before the harvest of summer maize in the 2021 growing season using the sampling method referenced in a previous study [16].

#### 2.3.2. Soil Physicochemical Properties Measurements

Soil compactness was expressed by soil bulk density and soil porosity. Soil bulk density was measured by dividing the weight of the dried soil by the soil volume. The calculation of soil porosity was calculated using the following formula:

$$P = \left(1 - \frac{d}{p}\right) \times 100\% \tag{1}$$

In this equation, P represents soil porosity, and d and *p* represent soil bulk density and the density of soil solid, respectively.

Soil aggregates were assessed using the wet sieve method modified by Wang et al. (2016) [17]. In this study, aggregates larger than 0.25 mm were defined as macro-aggregates. The mean weight diameter (MWD) was calculated using Equation (2), as described by the following formula:

$$MWD = \sum_{i=1}^{n} X_i W_i \tag{2}$$

where X$_i$ is the mean diameter of each size class and W$_i$ is the percentage of each size class with respect to the total sample.

The dichromate oxidation method was used to determine the SOC content. For the 0–20 cm soil layer, the SOC stock, sequestrated SOC ($\Delta SOC_{stock}$), and the annual increase rate of sequestrated SOC ($SOC_{SR}$) were calculated using the follow formulae:

$$SOC_{stock} = \sum_{i=1}^{n} SOC_i \times BD_i \times H_i \times 10 \tag{3}$$

$$\Delta SOC_{stock} = SOC_{stock-final} - SOC_{stock-intial} \tag{4}$$

$$SOC_{SR} = \frac{\Delta SOC_{stock}}{Years} \tag{5}$$

where $SOC_{stock}$ is the SOC stocks (Mg ha$^{-1}$), $SOC_i$ is the soil organic carbon content (g kg$^{-1}$), $BD_i$ is the soil bulk density (g cm$^{-3}$) at each soil layer, and $H_i$ is the soil depth (cm). $SOC_{stock-final}$ is the SOC stock under each treatment at harvest in 2021 and $SOC_{stock-intial}$ is the SOC stock before the experiment in 2019.

### 2.3.3. Soil Enzymatic Assay and Genetic Analysis of Soil Bacteria

Rhizosphere soil enzymes, including catalase, sucrase, urease, and acid phosphatase, were extracted using their corresponding detection kits provided by Suzhoug Grace Biotechnology Co., Ltd., China. Catalase, sucrase, urease, and acid phosphatase activities were measured using a FlexStation 3 MultiMode Microplate Reader (Molecular Devices LLC. 3860 N First Street San Jose, CA 95134, USA) at 240, 540, 630, and 660 nm, respectively. The genetic analysis of bacteria was conducted by Majorbio Bio-Pharm Technology Co. Ltd. (Shanghai, China).

### 2.3.4. Plant Dry Matter Accumulation and Grain Yield

Maize plant roots were collected using the digging method at the milk stage in the 2021 growing season, and three consecutive plants were chosen from the middle of each plot. Soil and root samples were collected from each of the following layers: 0–10, 10–20, 20–40, and 40–60 cm. After manual washing, root samples from each soil layer were selected and placed in a separate kraft paper bag and then oven-dried to calculate the root dry matter accumulation of each soil layer.

At physiological maturity in the 2021 growing season, five consecutive plants were chosen from the middle of each plot, and each plant was divided into stalk, leaves, sheath, tassel, bract, cob, and grains, which were oven-dried to calculate the DMA of each part and the total DMA. Harvest index = dry grain weight/above-ground dry matter weight. Moreover, all plants in a 4 m × 3 m area within the middle of each plot were manually harvested and the grain yield was calculated under the condition of 14% grain moisture content.

### 2.4. Data Analysis

The differences in soil physicochemical properties, soil enzyme activities, plant dry matter accumulation, and grain yield among the treatments were analyzed using SPSS 20.0 software. One-way analysis of variance (ANOVA) and Duncan's new multiple range test were used to test the significant difference at the $p < 0.05$ level. Graphs were constructed using either Microsoft Excel 2013 (Microsoft Corp., Redmond, WA, USA) software or SigmaPlot (ver. 12.0; Systat Software, San Jose, CA, USA). Pearson correlations were performed among various soil parameters, plant DMA, and maize grain yield.

## 3. Results

### 3.1. Percentage and Mean Weight Diameter (MWD) of Macro-Aggregates

The percentage and MWD of aggregates in response to the CAOFCF treatments differed between the soil layers (Figure 2). Compared to CF, the CAOFCF treatments obviously improved the percentage and MWD of aggregates in the 0–10 cm soil layer. Meanwhile, the percentages of aggregates of OFCF3 and OFCF4 were also markedly higher than in OFCF1 and OFCF2. However, the differences in MWD between OFCF1, OFCF2, OFCF3, and OFCF4 were not obvious, except that OFCF4 was significantly lower than

OFCF2. In the 10–20 cm soil profile, the percentage differences in macro-aggregates between CF, OFCF1, OFCF3, and OFCF4 were not obvious, but all were significantly higher than OFCF2; no obvious difference was found for MWD among all treatments.

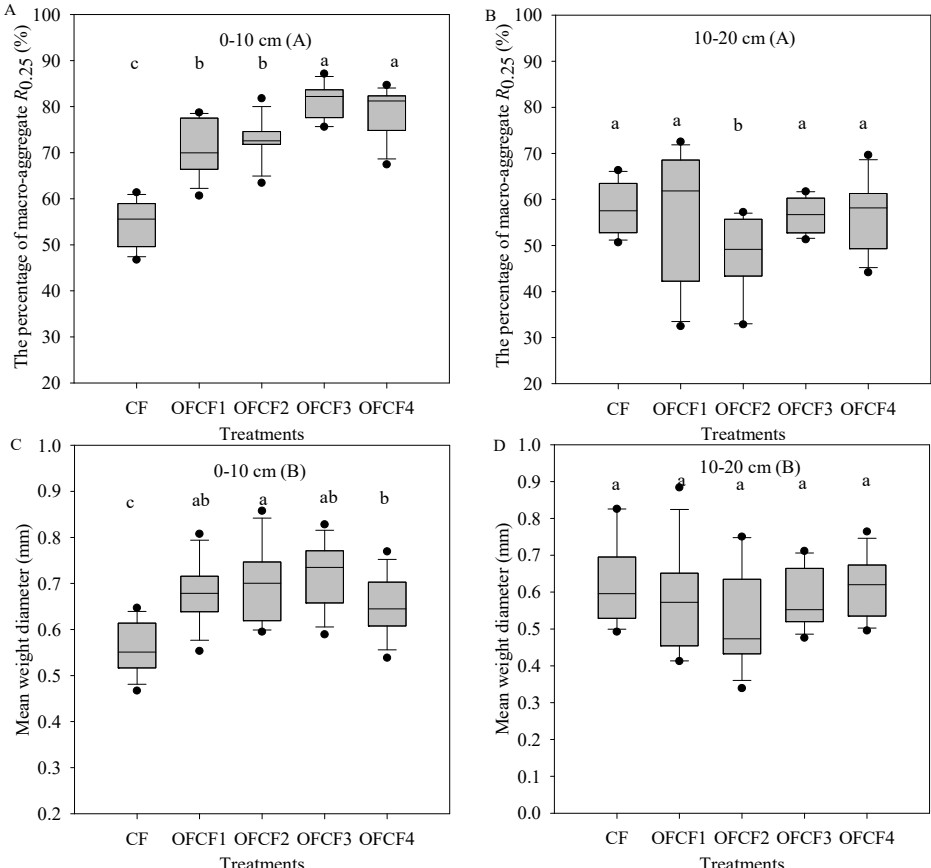

**Figure 2.** Effect of CAOFCF on the percentage of macro-aggregates (**A**) and MWD (**B**) in the 0–20-cm soil layer. (**C**) in the 0–10 cm soil layer, (**D**) 10–20 cm soil profile. Different letters indicate significant differences among treatments at $p < 0.05$.

### 3.2. Soil Bulk Density and Soil Porosity

As shown in Table 1, the PCFSOF treatments significantly affected the soil bulk density and porosity in the topsoil layer. Compared to CF, the OFCF1, OFCF2, OFCF3, and OFCF4 treatments reduced the soil bulk density in the 0–20-cm soil layer on average by 13.5%, 12.2%, 12.2%, and 12.2%, respectively ($p < 0.05$). In contrast, the OFCF1, OFCF2, OFCF3, and OFCF4 treatments significantly increased the soil porosity by 20.2%, 18.3%, 17.4%, and 18.1%, respectively ($p < 0.05$).

**Table 1.** Effect of PCFSOF on soil bulk density and soil porosity of topsoil.

| Treatments | Soil Bulk Density (g cm−3) | | Soil Porosity (%) | |
|---|---|---|---|---|
| | 0–10 cm | 10–20 cm | 0–10 cm | 10–20 cm |
| CF | 1.38 ± 0.03 [a] | 1.48 ± 0.03 [a] | 36.40 ± 1.28 [b] | 31.48 ± 1.32 [b] |
| OFCF1 | 1.27 ± 0.05 [ab] | 1.29 ± 0.04 [b] | 41.28 ± 2.31 [ab] | 40.31 ± 1.93 [a] |
| OFCF2 | 1.27 ± 0.03 [ab] | 1.32 ± 0.02 [b] | 41.43 ± 1.44 [ab] | 38.86 ± 1.04 [a] |
| OFCF3 | 1.26 ± 0.03 [b] | 1.34 ± 0.01 [b] | 41.77 ± 1.49 [a] | 37.95 ± 0.56 [a] |
| OFCF4 | 1.24 ± 0.0.02 [b] | 1.35 ± 0.02 [b] | 42.58 ± 1.02 [a] | 37.59 ± 0.92 [a] |

Note: Values are the mean ± standard deviation. Different lowercase letters within a column indicate significant differences among treatments ($p < 0.05$).



### 3.3. SOC Content, Storage, and Sequestration

The CAOFCF treatments markedly improved the SOC content in the 0–10 cm soil layer compared to the CF treatment ($p < 0.05$); however, no difference was observed in SOC content at the depth of the 10–20 cm soil layer among treatments (Figure 3). Compared to CF, OFCF1, OFCF2, OFCF3, and OFCF4 increased the SOC content in the 0–10 cm soil layer by 13.3%, 14.2%, 18.0%, and 21.6%, respectively.

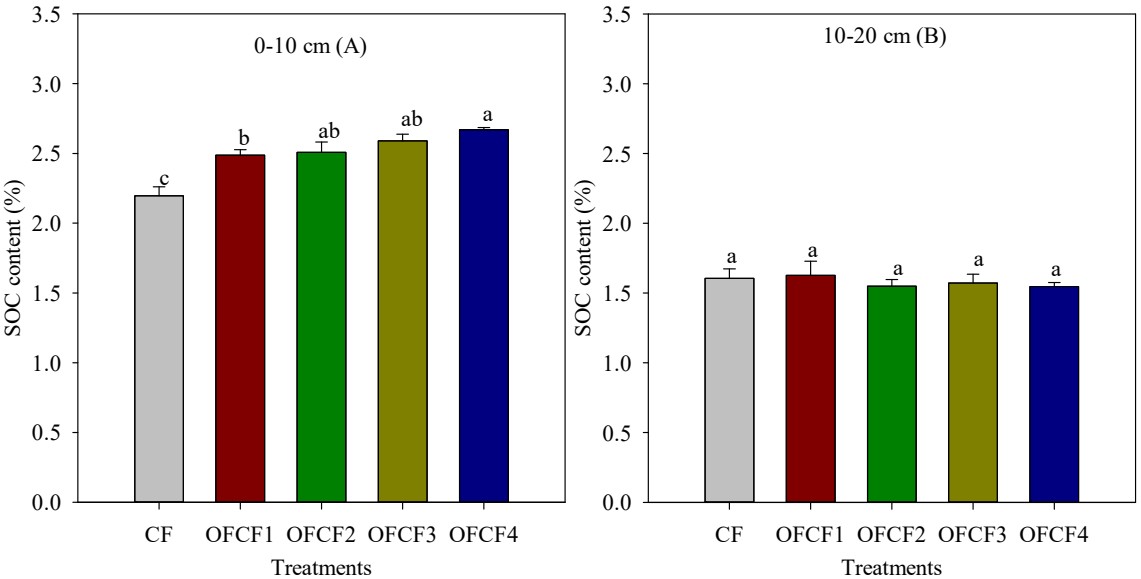

**Figure 3.** Effect of CAOFCF on SOC content in the 0–10 cm (**A**) and 10-20 cm (**B**) topsoil. Different letters indicate significant differences among treatments at $p < 0.05$.

The SOC stock in the 0–20 cm soil layer at harvest in 2021 ranged from 51.71–54.70 Mg ha$^{-1}$ for all treatments with the following order: OFCF4 > OFCF3 > OFCF2 = OFCF1 > CF (Table 2). Compared to the topsoil SOC stock before the sowing of summer maize in 2019, a net increase in SOC stock was observed for the CAOFCF treatments, but there was a loss for the CF treatment. Among all treatments, OFCF4 had the highest amount and rate of SOC sequestration in the topsoil layer, and the difference in SOC stock change between OFCF1 and OFCF2 was not obvious.

**Table 2.** Effect of CAOFCF on SOC stock and sequestration in the topsoil.

| Treatments | SOC Stock in 2019 (Mg ha$^{-1}$) | SOC Stock in 2021 (Mg ha$^{-1}$) | Sequestration Carbon (Mg ha$^{-1}$) | Sequestrated Carbon Rate (Mg ha$^{-1}$ yr$^{-1}$) |
|---|---|---|---|---|
| CF | 52.32 ± 0.38 | 51.71 ± 0.22 | −0.61 ± 0.22 | −0.20 ± 0.24 |
| OFCF1 | 52.32 ± 0.38 | 52.74 ± 1.03 | 0.42 ± 1.03 | 0.14 ± 0.34 |
| OFCF2 | 52.32 ± 0.38 | 52.74 ± 1.25 | 0.42 ± 1.25 | 0.14 ± 0.42 |
| OFCF3 | 52.32 ± 0.38 | 54.20 ± 0.58 | 1.88 ± 0.58 | 0.63 ± 0.19 |
| OFCF4 | 52.32 ± 0.38 | 54.70 ± 0.19 | 2.38 ± 0.19 | 0.79 ± 0.06 |

### 3.4. Soil Enzymatic Activities

The responses of soil enzymatic activities to the CAOFCF treatments differed (Figure 4). The urease activity in the OFCF2 and OFCF3 treatments was significantly higher than that in the CF treatment and increased by 8.6% and 14.8%, respectively. However, OFCF4 significantly reduced the urease activity compared to CF. There was no obvious difference in the sucrase activity among all treatments, except for OFCF4, which was significantly lower than OFCF1 and OFCF2. As the application rate of organic fertilizer increased, the catalase activity decreased, and OFCF3 and OFCF4 were significantly lower than CF, OFCF1, and OFCF2. Compared to CF, OFCF1, OFCF2, OFCF3, and OFCF4 increased the

acid phosphatase activity by 5.3%, 14.9%, 24.3%, and 13.5%, respectively, and significant differences were observed for the OFCF2, OFCF3, and OFCF4 treatments.

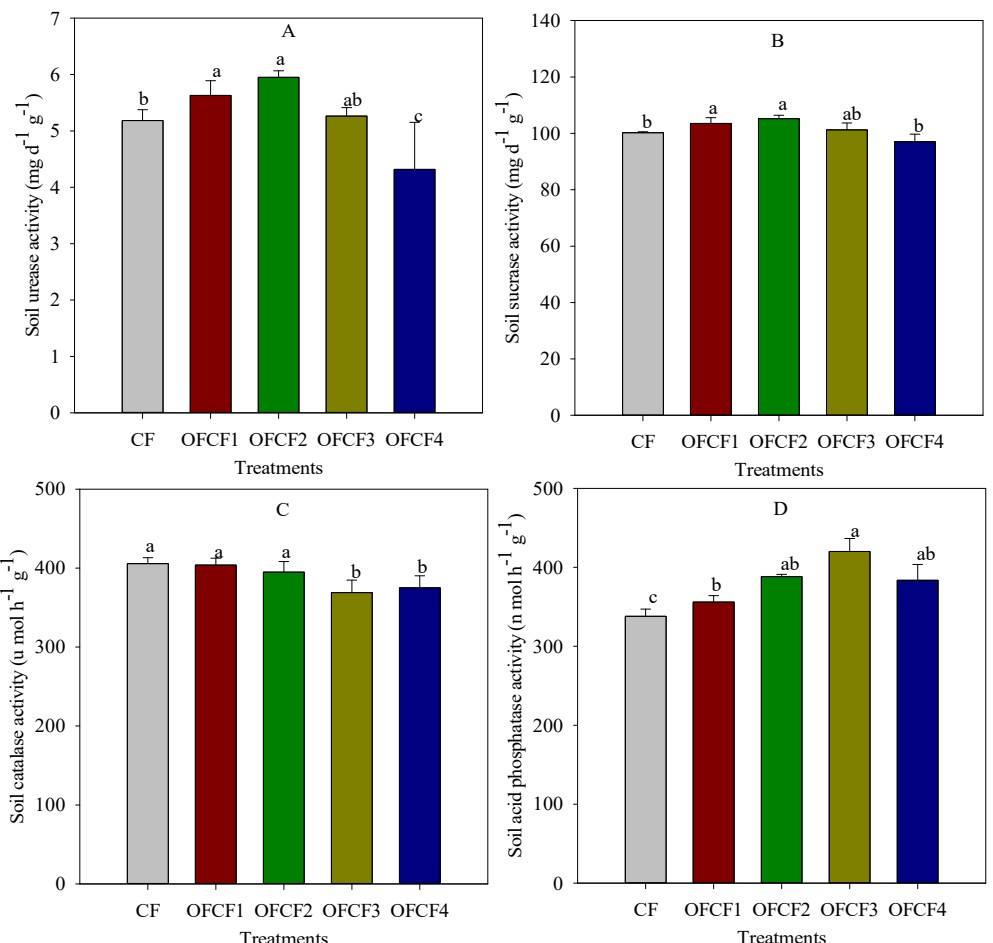

**Figure 4.** Effect of PCFSCF treatments on urease (**A**), sucrase (**B**), catalase (**C**), and acid phosphatase (**D**) activities of rhizosphere soil. Vertical bars indicate the standard error. Different letters above the columns indicate significant differences among treatments at $p < 0.05$.

### 3.5. Soil Microbial Diversity and Community Composition

In this study, the Shannon index was used to reflect the alpha diversity (OTU-level) of the soil bacteria. The Shannon index values of CF, OFCF1, OFCF2, OFCF3, and OFCF4 were 6.599, 6.626, 6.583, 6.551, and 6.595, respectively (Figure 5a). No obvious difference in Shannon index was observed among the treatments, with the exception of OFCF3, which was significantly lower than OFCF2. PCoA analysis showed that the community composition of the soil bacteria was changed by the CAOFCF treatments, especially for OFCF3 and OFCF4 (Figure 5b). The first and second principal coordinates explained 32.98% and 7.46%, respectively, of the differences among the five treatments (Figure 5b).

At the phylum level, the taxonomic analysis showed that the dominant phyla were *Proteobacteria*, *Actinobacteria*, *Acidobacteria*, *Chloroflexi*, *Planctomycetota*, and *Bacteroidota* in the surface rhizosphere soil, accounting for over 80% of the bacterial community (Figure 6). Among the top six phyla, the CAOFCF treatments improved the relative abundance of *Actinobacteria* and *Chloroflexi*, especially for the OFCF1, OFCF3, and OFCF4 treatments. However, the relative abundance of *Proteobacteria* and *Planctomycetota* under the CAOFCF treatments was reduced, and a significant difference was observed for *Planctomycetota*.

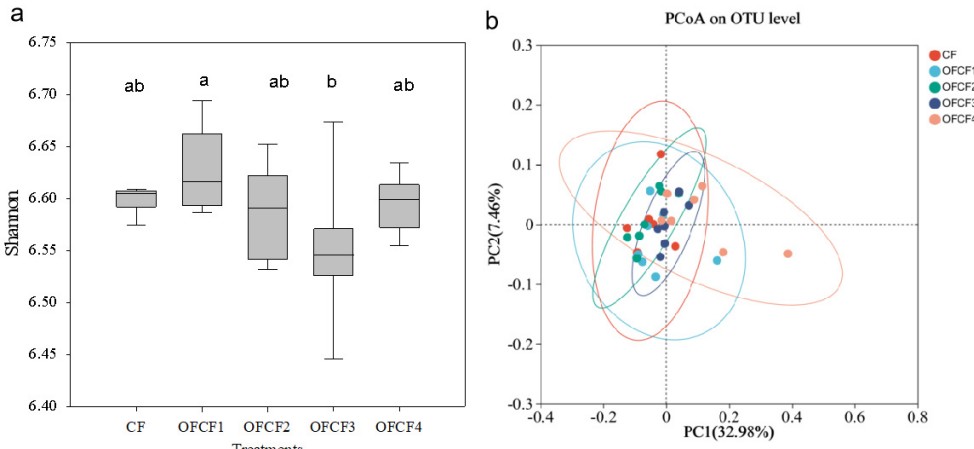

**Figure 5.** Effect of CAOFCF treatments on bacterial alpha diversity (**a**) and beta diversity (**b**). Different letters above the columns indicate significant differences among treatments at $p < 0.05$.

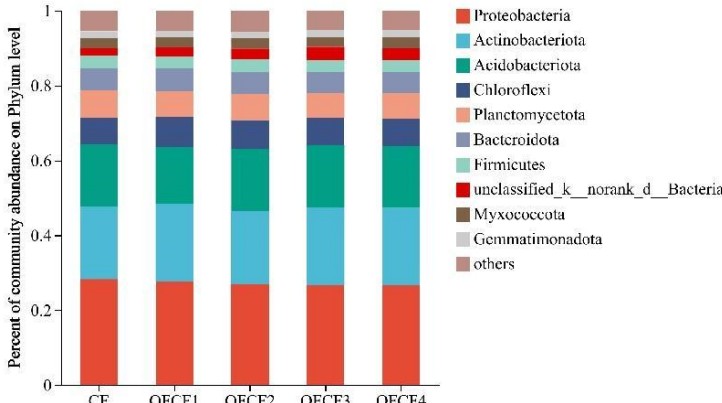

**Figure 6.** The relative community abundance of bacteria at the phylum level under the different treatments.

OFCF1, OFCF3, and OFCF4 increased the bacterial functional abundance compared to CK, especially for the OFCF3 and OFCF4 treatments (Figure 7). However, analysis of the relative abundance of functional genes shown in the COG classification histogram revealed that OFCF1 was associated with enriched amino acid transport and metabolism as well as carbohydrate transport and metabolism.

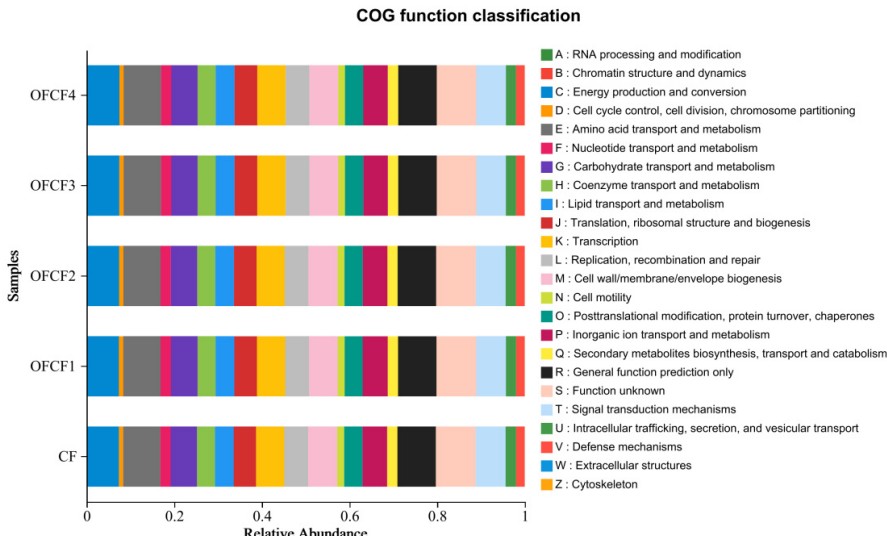

**Figure 7.** COG function classification histogram for the different treatments.



### 3.6. Plant DMA, Grain Yield, and Harvest Index

As shown in Figure 8, the root and above-ground DMA, grain yield, and harvest index were significantly affected by the CAOFCF treatments ($p < 0.05$). As compared to the CF treatment, OFCF1 and OFCF2 significantly increased the root dry matter accumulation in the 0–10 and 10–20 cm soil depths. Similarly, OFCF1 and OFCF2 increased the above-ground DMA by 16.3% and 8.7% respectively, compared to that of CF. The grain yields of OFCF1, OFCF2, and OFCF3 were 13.1%, 8.9%, and 7.8% higher than that of CF. Although no obvious difference in the harvest index was observed among CF, OFCF1, OFCF2, and OFCF3, the harvest index of OFCF4 was significantly lower than that of OFCF1 and OFCF2.

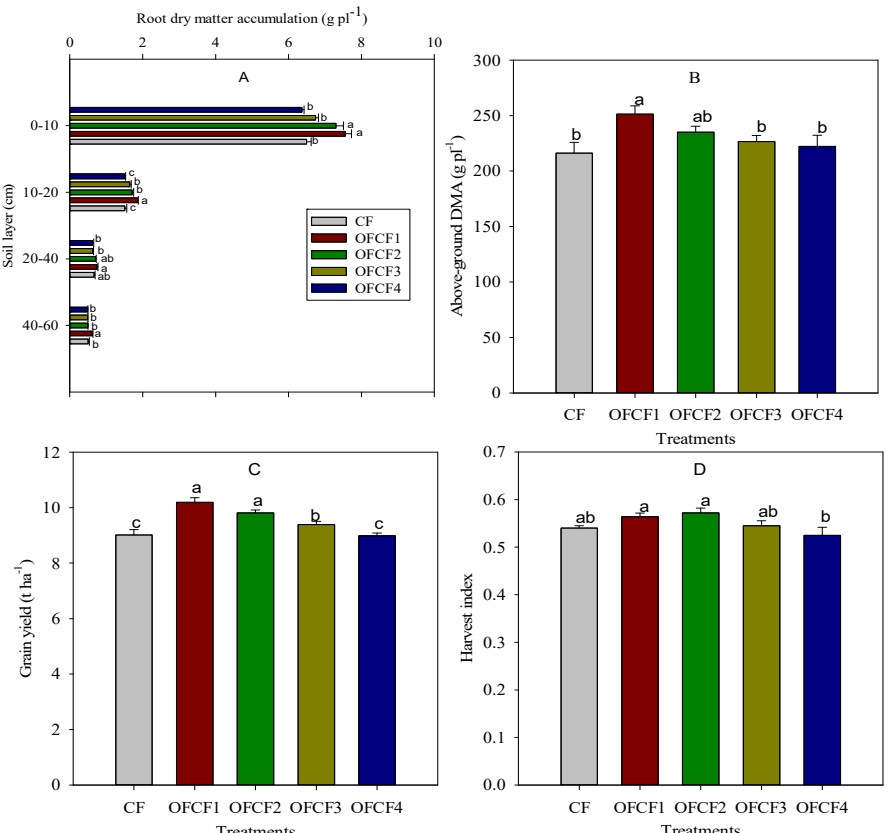

**Figure 8.** Effect of CAOFCF on root DMA (**A**), above-ground DMA (**B**), grain yield (**C**), and harvest index (**D**) of summer maize. Different small letters above the columns indicate significant differences at $p < 0.05$.

### 3.7. Relationship among Soil Indicators and Grain Yield

The correlations among the soil indicators (including SOC, soil bulk density, macroaggregate, and soil enzyme activities) and grain yield of summer maize are presented in Figure 9. For all the soil indicators measured in the 0–20 cm soil layer, the SOC was significantly negatively correlated with the soil bulk density and positively correlated with the proportion of macro-aggregates and the mean weight diameter. Moreover, the soil urease and sucrase activities were also positively correlated. The maize grain yield had no direct relationship with various soil indicators in the 0–20 cm soil layer. The BD and MWD were negatively correlated, whereas strong positive correlations were observed between SOC and R0.25, urease and sucrase activities, sucrase activity and root DMA, root DMA and above-ground DMA, and aboveground DMA and grain yield.

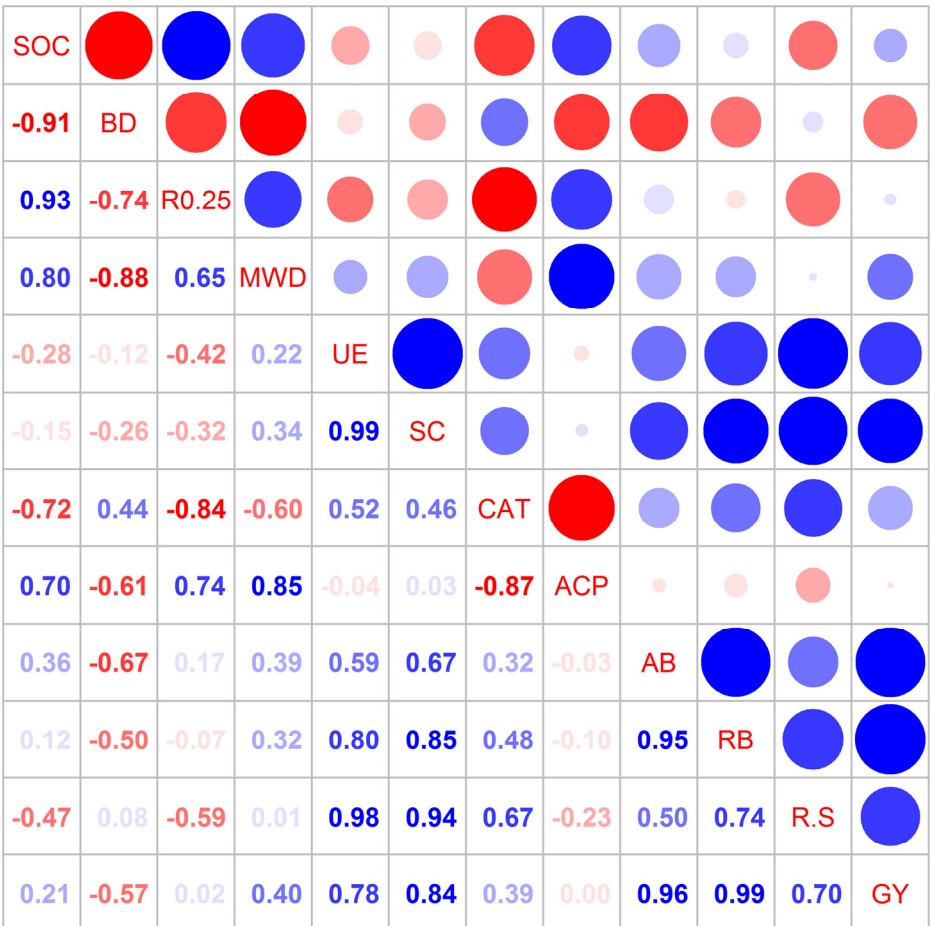

**Figure 9.** Pearson's correlation analysis of soil quality parameters and maize yield as affected by different treatments for 0–20 cm depth. The color and number denote 0.25 the magnitude of the relationship. SOC, soil organic content; BD, bulk density; R0.25, proportion of macro-aggregates with a diameter > 0.25 mm; MWD, mean weight diameter; UE, SC, CAT, and ACP represent the soil activities of urease, sucrase, catalase, and acid phosphatase, respectively; AB, above-ground biomass; RB, root biomass; RS, root to shoot ratio; GY, grain yield.

## 4. Discussion

### 4.1. Short-Term Responses of Topsoil Structure and SOC Sequestration to CAOFCF Treatments

Soil aggregate stability can be used as an important index to evaluate the stabilization and degradation of soil texture [6,18–20]. Some studies have shown that increased soil aggregate stability was associated with organic fertilizer input in agricultural ecosystems [21]. In addition, previous studies have demonstrated that co-application of organic-inorganic fertilizer had a positive role in soil macro-aggregate stability [22,23]. However, most of these results were obtained from long-term experiments, whereas little information has focused on the short-term responses of topsoil aggregate stability to CAOFCF treatment. In this study, CAOFCF significantly increased the percentage of macro-aggregates in the 0–10 cm soil layer; however, this positive effect was not obvious for the 10–20 cm soil layer (Figure 2). These results suggested that short-term CAOFCF was beneficial to soil aggregate stability in the 0–10 cm soil layer, which was consistent with a previous study [24]. Given that organic fertilizer contains binding agents, this may explain the improved stability of soil aggregates [25]. Soil MWD is another indicator that reflects soil aggregate stability [26]. Consistent with the macro-aggregates, CAOFCF significantly improved the MWD in the 0–10 cm soil layer compared to CF. The higher MWD under the CAOFCF treatments indicated the stronger stability of aggregates and anti-erosion ability of the topsoil.

Soil bulk density and porosity are considered important parameters that reflect soil structure [27,28]. Numerous studies have reported that soil bulk density could be reduced by applying organic fertilizer long-term [29,30]. Nevertheless opposite results were also observed [31]. Compared to CF, the present results confirmed that all CAOFCF treatments decreased the soil bulk density of the topsoil layer (Table 1). However, no significant difference was found among the CAOFCF treatments. The results clearly revealed that short-term CAOFCF can prevent the occurrence of soil compaction. Since organic materials are characterized by low bulk density and higher porosity [32], they also play a decisive role in binding soil particles and resistance against the formation of compacted soil. Therefore, the application of organic matter is beneficial in changing the topsoil physical properties [33].

Promoting SOC sequestration is an important measure for maintaining and restoring soil quality, which is beneficial to sustainable agronomic productivity [34]. The amount of SOC sequestration is affected by the quantity and quality of exogenous organic matter input [35]. In our study, three years of continuous CAOFCF markedly increased the SOC stock in the topsoil layer compared to CF, but significant differences were only observed for the OFCF3 and OFCF4 treatments. Our results implied that applying a larger amount of organic fertilizer (i.e., at least 4.08 Mg ha$^{-1}$) each year can increase the SOC sequestration of the topsoil in the short term (three years), which was consistent with previous results [36]. Mandal et al. (2020) reported that the increase in SOC content contributes a lot to farmland SOC sequestration [37]. As is well known, organic fertilizer is an important SOC source [38–40]; thus, the higher SOC sequestration observed under OFCF3 and OFCF4 may be closely related to the improved SOC content in the topsoil layer (Figure 3). However, applying a lower amount of organic fertilizer obviously makes it difficult to increase SOC sequestration in the short term. Generally, improved SOC sequestration was observed in long-term field experiments [11,41,42]. The present study revealed that a significant increase in topsoil SOC sequestration can also be realized by applying a larger amount of organic fertilizer in the short term.

### 4.2. Short-Term Responses of Rhizosphere Soil Enzyme Activities and Microbal Diversity to CAOFCF Treatments

Generally, soil nutrient cycling and metabolic processes are associated with various soil enzymes [43,44]. Previous studies confirmed that soil enzyme activities change when organic fertilizer is substituted with chemical fertilizer [5,10–12]. For example, Ren et al. (2021) demonstrated that a higher substitution rate of chemical fertilizer with organic fertilizer significantly reduced soil catalase activities [4], which corresponded well with our observations. However, the present results suggested that the highest soil urease and sucrase activities were obtained in the OFCF2 treatment (i.e., 30% substitution of chemical fertilizer by organic fertilizer) (Figure 4), which differed from a previous report [5] in which higher urease activity was associated with a higher organic fertilizer substitution rate. This discrepancy might be attributed to the length of the trials in these studies. Moreover, the acid phosphatase activity increased as the organic substitution rate increased. Altogether, these findings indicated that short-term co-application of 30% organic fertilizer and 70% chemical fertilizer can simultaneously increase the urease, sucrase, and phosphatase activities in maize rhizosphere soil, which benefits soil fertility and nutrient cycling.

Soil microbial diversity and community composition reflect soil fertility and quality [45,46]. It is well known that fertilization can influence soil microbial diversity [47–49]. A previous meta-analysis showed that organic fertilizer application improved the soil microbial diversity [50]. Compared to CF, no obvious differences in bacterial alpha diversity and beta diversity were observed for the CAOFCF treatments. Our study suggested no obvious difference in soil bacterial diversity was induced by the tested treatments, which differed from the results by Ren et al. (2021) [4]. The present study revealed that co-application of organic-inorganic fertilizer did not improve the topsoil bacterial diversity, suggesting that significant effects may be associated with trial length, organic fertilizer type, soil texture,

and field cropping system [1,51,52]. In our study, *Proteobacteria*, *Actinobacteria*, *Acidobacteria*, *Chloroflexi*, *Planctomycetota*, and *Bacteroidota* were the top six dominant bacterial phyla, and applying organic fertilizer markedly improved the relative richness of *Actinobacteria* and *Chloroflexi* (Figure 6). It is generally known that *Actinobacteria* is an important microbial group that plays an essential role in natural ecosystem carbon and N cycles [53]. Improved *Actinobacteria* richness in the rhizosphere soil definitely promotes nutrient cycling and soil quality [54]. As a dominant bacterial phyla, *Chloroflexi* not only plays a part in driving biogeochemical cycles [55], but it also affects the soil's multifunctional resistance [56]. Therefore, the increased relative abundance of *Chloroflexi* under the CAOFCF treatments was beneficial to biogeochemical cycles and soil resistance to abiotic stress. Compared to CF, OFCF1 increased the relative richness of functional genes associated with amino acid transport and metabolism as well as carbohydrate transport and metabolism, which implied that short-term 15–30% substitution of chemical fertilizer with organic fertilizer with an equal N rate was conducive to rhizosphere soil carbon and nitrogen metabolism rather than a higher application amount.

*4.3. Effect of Short-Term CAOFCF on Summer Maize Productivity*

A growing body of evidence has shown that CAOFCF is conducive to crop productivity [1,57–59]. However, few studies have focused on the short-term effect of CAOFCF on maize productivity. In the present study, the 3-year trial showed that the grain yields in the OFCF1 and OFCF2 treatments were obviously higher than that in CF, suggesting that short-term CAOFCF with a lower amount of organic fertilizer (i.e., 1.36–2.72 Mg ha$^{-1}$) was beneficial to maize grain yield in a medium-productivity cinnamon soil, which was similar to a previous report [9]. However, the larger application amount of organic fertilizer did not contribute to the grain yield in the short term. Given that organic fertilizer is characterized by slow nutrient release [60], co-application of a larger amount of organic fertilizer with a lower amount of chemical fertilizer in the short term may not meet the nutrient requirements of crop plants during the whole growth period, which will lead to an adverse effect on crop productivity due to insufficient available nutrients.

A number of studies have demonstrated that the changes in topsoil physicochemical properties caused by organic matter amendment are associated with maize yield [4,34]. However, our correlation analysis showed that maize grain yield had no obvious direct relationship with topsoil properties, including SOC, soil bulk density, soil aggregate stability, and soil enzyme activities. A possible explanation for this difference is the different trial lengths and organic materials applied in these studies. Nevertheless, the correlation analysis revealed that the activities of some key enzymes in the rhizosphere soil (i.e., urease and sucrase) were directly correlated with the root-shoot ratio and root dry weight. Meanwhile, the root dry weight and grain yield were positively correlated (Figure 9). The present results revealed that the higher productivity observed under OFCF1 and OFCF2 was related to a stronger root system, which was mainly driven by the improved rhizosphere soil urease and sucrase activities. Usually, soil urease and sucrase are pivotal in soil nutrient cycling [61–63], and crop root has been associated with these enzymes in fields with different soil textures [64,65]. In brief, compared to the fertilization practice used by most local smallholder farmers (i.e., applying chemical fertilizer alone), replacing 15–30% chemical fertilizer with organic fertilizer with an equal N rate could be recommended as an optimized fertilization practice since this fertilization practice not only significantly increased the productivity of summer maize, but it also improved the topsoil quality, which is feasible to developing sustainable agriculture in a medium-productivity meadow-cinnamon soil.

**5. Conclusions**

The present study revealed that 3 years of CAOFCF was beneficial to the topsoil physicochemical properties by regulating soil aggregate stability, bulk density, and SOC sequestration. Compared to CF, CAOFCF did not increase the soil bacterial diversity, but it improved the relative richness of *Actinobacteria* and *Chloroflexi* as well as the bacterial

functional abundance. Moreover, OFCF1 and OFCF2 significantly increased the soil urease, sucrase, and acid phosphatase activities, as well as the root dry weight, shoot dry weight, and grain yield of maize. The correlation analysis implied that the improved root system under the CAOFCF treatments was closely related to the higher productivity, which was driven by the activated soil urease and sucrase. Overall, short-term CAOFCF treatment (i.e., replacing 15–30% chemical fertilizer with organic fertilizer with an equal N rate) can improve topsoil properties and maize productivity in a medium-productivity meadow-cinnamon soil, which can contribute to sustainable agriculture development.

**Author Contributions:** Funding acquisition and Conceptualization, L.Z. (Lichao Zhai), M.Z. and L.Z. (Lihua Zhang); data collection and writing—original draft, L.Z. (Lichao Zhai); writing—review and editing, L.Z. (Lichao Zhai), M.Z., J.C., J.Z. and X.J. All authors have read and agreed to the published version of the manuscript.

**Funding:** This study was financially supported by the HAAFS Science and Technology Innovation Special Project (2022KJCXZX-LYS-9), the Key Research and Development Program of Hebei Province (20326401D), the Natural Science Foundation of Hebei Province (C2021301004), and the Talents Construction Project of Science and Technology Innovation, Hebei Academy of Agriculture and Forestry Sciences (C19R02-1-1).

**Data Availability Statement:** The data presented in this study are available on request from the corresponding author.

**Conflicts of Interest:** The authors declare no conflict of interest.

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
