# Peer review of "Short-Term Co-Application of Organic and Chemical Fertilizer Benefits Topsoil Properties and Maize Productivity in a Medium-Productivity Meadow-Cinnamon Soil"

_agronomy, doi:10.3390/agronomy13030944_

Round 1
Reviewer 1 Report
Dear author;
The present manuscript aims to investigate the Effect of short-term combined application of organic fertilizer and chemical fertilizer on topsoil properties and productivity of summer maize in a medium-productivity meadow-cinnamon soil. I believe that the following paper might be considered for publication in the journal of agronomy after the proposed revision.
In general, the research deals with an important topic that will be a plus in the effect of fertilization on the rhizosphere.
There are some small English writing errors that need to be modified and corrected, the authors should check the whole manuscript and correct them.
Any amount of plagiarism is unacceptable in academic writing. Authors should aim for zero percent plagiarism in their papers (I found more than 40% in your paper), and any detected instances of plagiarism should be addressed and corrected. It is important to ensure that all sources are properly cited and that any direct quotes are indicated as such. Generally, it is best to use one's own words and ideas when writing and attribute all sources appropriately.
Find attached the report of plagiarism to address all the issues.
ABSTRACT:
The abstract provides a clear overview of a 3-year field trial examining the effects of combined organic and chemical fertilizers on soil properties and maize productivity. However, there are a few issues with the abstract:
While the abstract describes the research design and key findings, it is not entirely clear what the research question was. The abstract could benefit from a more focused and concise statement of the research question.
Overuse of acronyms: The abstract includes several acronyms, such as CF, OFCF, SOC, and DMA, without providing adequate explanation or context. This can make the abstract difficult to follow for readers who are not familiar with the terminology.
Introduction:
The introduction is good and clearly outlines the study's problematic and objectives.
Material and methods:
Line 79 : "Field trail" should be "field trial."
In line 80: do you mean the Northern Huang–Huai–Ha? If yes; pleas change HHH to the full location name.
Line 82: "ËšC" should be replaced with the degree symbol (°C).
Line 84-86: I would rather prefer to rewrite the phrase as follow: Nutrient content in the top 0-20 cm soil layer before the experiment was set up in 2019 comprised 1.87% organic matter, 1.04 g/kg total N, 80 mg/kg available N, 21.4 mg/kg available P, and 113.9 mg/kg available K."
The references should follow the theme of the journal
find attached the plagiarism reports so that you can improve your paper.

Author Response
Responses to Reviewer 1’ Comments
Comment 1: There are some small English writing errors that need to be modified and corrected, the authors should check the whole manuscript and correct them.
Answer: Thank you for your suggestion. I myself and a person proficient in professional English have revised the English grammar of this manuscript in the revised manuscript.
Comment 2: Any amount of plagiarism is unacceptable in academic writing. Authors should aim for zero percent plagiarism in their papers (I found more than 40% in your paper), and any detected instances of plagiarism should be addressed and corrected. It is important to ensure that all sources are properly cited and that any direct quotes are indicated as such. Generally, it is best to use one's own words and ideas when writing and attribute all sources appropriately. Find attached the report of plagiarism to address all the issues.
Answer: We agree with your comment on any amount of plagiarism is unacceptable in academic writing. However, we don’t quite agree with the higher plagiarism rate reported (more than 40%). First, the institutional name of authors are fixed, it should not be in the scope of review of papers. Second, some academic terms, just as ‘dry matter accumulation’, ‘soil organic carbon’, ‘organic fertilizer’ et al., are commonly used internationally, so we don’t think it is a plagiarism. Third, some Figures (Figure 2 , 4, and 8) were labeled as a plagiarism in the review report, we strongly oppose it, because these Figures are made according to our data, and these data has never been used or published in anywhere. Fourth, the title, abstract, introduction, result, and conclusion sections of this manuscript were all constructed and written in our own words, let’s take the abstract section for example, it is all written by ourselves, but I don’t why almost all of it is labeled as a plagiarism. Furthermore, a similar situation also exists in the discussion section (4.2 and 4.3). Therefore, we don’t this plagiarism report can reflect the real paper review. Even so, we have done our best to reduce the plagiarism rate showed in the report and improved the text in the revised manuscript.
ABSTRACT:
The abstract provides a clear overview of a 3-year field trial examining the effects of combined organic and chemical fertilizers on soil properties and maize productivity. However, there are a few issues with the abstract:
Comment 3:While the abstract describes the research design and key findings, it is not entirely clear what the research question was. The abstract could benefit from a more focused and concise statement of the research question.
Answer: Thank you for your suggestion. Actually, the research question is raised in the abstract section. In Line 13-15, we have mentioned “it remains unknown whether short-term CAOFCF can improve soil properties and maize productivity simultaneously in a medium-productivity meadow-cinnamon soil”. It is actually the main research question in our study. In order to make it more clear for readers, we have made a further revision in the revised manuscript (Line 13-15 in the revised MS).
Comment 4: Overuse of acronyms: The abstract includes several acronyms, such as CF, OFCF, SOC, and DMA, without providing adequate explanation or context. This can make the abstract difficult to follow for readers who are not familiar with the terminology.
Answer: Thank you for your suggestion. First, CF, OFCF1-4, is representing the five treatments in our study, it is not abbreviations of any phrase. Second, some abbreviations used in the abstract and other part of this manuscript, just as SOC, DMA et al., were spelled in full when it was first mentioned in the revised manuscript.( Line 22, 27 in the revised manuscript)
Material and Methods section:
Comment 5: Line 79 : "Field trail" should be "field trial."
Answer: Thank you for your suggestion, we have made the specific change in the revised manuscript (Line 83 in the revised manuscript).
Comment 6: In line 80: do you mean the Northern Huang–Huai–Ha? If yes; pleas change HHH to the full location name.
Answer: We agree with this comment, and have made the specific change in the revised manuscript(Line 84 in the revised manuscript).
Comment 7: Line 82: "ËšC" should be replaced with the degree symbol (°C).
Answer: We agree with this comment, and have made the specific change in the revised manuscript (Line 87 in the revised manuscript).
Comment 8: Line 84-86: I would rather prefer to rewrite the phrase as follow: Nutrient content in the top 0-20 cm soil layer before the experiment was set up in 2019 comprised 1.87% organic matter, 1.04 g/kg total N, 80 mg/kg available N, 21.4 mg/kg available P, and 113.9 mg/kg available K."
Answer: We agree with this comment, and have rewrite this sentence to make it more clear for readers in the revised manuscript (Line 87-90 in the revised manuscript).
Comment 9: The references should follow the theme of the journal.
Answer: Thank you for your suggestion. All the references in this manuscript have been checked, and we have made a revision on it according to format requirements of Agronomy-Basel journal (Reference section in the revised manuscript).
Responses to Reviewer 1’ Comments
Comment 1: There are some small English writing errors that need to be modified and corrected, the authors should check the whole manuscript and correct them.
Answer: Thank you for your suggestion. I myself and a person proficient in professional English have revised the English grammar of this manuscript in the revised manuscript.
Comment 2: Any amount of plagiarism is unacceptable in academic writing. Authors should aim for zero percent plagiarism in their papers (I found more than 40% in your paper), and any detected instances of plagiarism should be addressed and corrected. It is important to ensure that all sources are properly cited and that any direct quotes are indicated as such. Generally, it is best to use one's own words and ideas when writing and attribute all sources appropriately. Find attached the report of plagiarism to address all the issues.
Answer: We agree with your comment on any amount of plagiarism is unacceptable in academic writing. However, we don’t quite agree with the higher plagiarism rate reported (more than 40%). First, the institutional name of authors are fixed, it should not be in the scope of review of papers. Second, some academic terms, just as ‘dry matter accumulation’, ‘soil organic carbon’, ‘organic fertilizer’ et al., are commonly used internationally, so we don’t think it is a plagiarism. Third, some Figures (Figure 2 , 4, and 8) were labeled as a plagiarism in the review report, we strongly oppose it, because these Figures are made according to our data, and these data has never been used or published in anywhere. Fourth, the title, abstract, introduction, result, and conclusion sections of this manuscript were all constructed and written in our own words, let’s take the abstract section for example, it is all written by ourselves, but I don’t why almost all of it is labeled as a plagiarism. Furthermore, a similar situation also exists in the discussion section (4.2 and 4.3). Therefore, we don’t this plagiarism report can reflect the real paper review. Even so, we have done our best to reduce the plagiarism rate showed in the report and improved the text in the revised manuscript.
ABSTRACT:
The abstract provides a clear overview of a 3-year field trial examining the effects of combined organic and chemical fertilizers on soil properties and maize productivity. However, there are a few issues with the abstract:
Comment 3:While the abstract describes the research design and key findings, it is not entirely clear what the research question was. The abstract could benefit from a more focused and concise statement of the research question.
Answer: Thank you for your suggestion. Actually, the research question is raised in the abstract section. In Line 13-15, we have mentioned “it remains unknown whether short-term CAOFCF can improve soil properties and maize productivity simultaneously in a medium-productivity meadow-cinnamon soil”. It is actually the main research question in our study. In order to make it more clear for readers, we have made a further revision in the revised manuscript (Line 13-15 in the revised MS).
Comment 4: Overuse of acronyms: The abstract includes several acronyms, such as CF, OFCF, SOC, and DMA, without providing adequate explanation or context. This can make the abstract difficult to follow for readers who are not familiar with the terminology.
Answer: Thank you for your suggestion. First, CF, OFCF1-4, is representing the five treatments in our study, it is not abbreviations of any phrase. Second, some abbreviations used in the abstract and other part of this manuscript, just as SOC, DMA et al., were spelled in full when it was first mentioned in the revised manuscript.( Line 22, 27 in the revised manuscript)
Material and Methods section:
Comment 5: Line 79 : "Field trail" should be "field trial."
Answer: Thank you for your suggestion, we have made the specific change in the revised manuscript (Line 83 in the revised manuscript).
Comment 6: In line 80: do you mean the Northern Huang–Huai–Ha? If yes; pleas change HHH to the full location name.
Answer: We agree with this comment, and have made the specific change in the revised manuscript(Line 84 in the revised manuscript).
Comment 7: Line 82: "ËšC" should be replaced with the degree symbol (°C).
Answer: We agree with this comment, and have made the specific change in the revised manuscript (Line 87 in the revised manuscript).
Comment 8: Line 84-86: I would rather prefer to rewrite the phrase as follow: Nutrient content in the top 0-20 cm soil layer before the experiment was set up in 2019 comprised 1.87% organic matter, 1.04 g/kg total N, 80 mg/kg available N, 21.4 mg/kg available P, and 113.9 mg/kg available K."
Answer: We agree with this comment, and have rewrite this sentence to make it more clear for readers in the revised manuscript (Line 87-90 in the revised manuscript).
Comment 9: The references should follow the theme of the journal.
Answer: Thank you for your suggestion. All the references in this manuscript have been checked, and we have made a revision on it according to format requirements of Agronomy-Basel journal (Reference section in the revised manuscript).

Reviewer 2 Report
The contribution is thematically focused on the issue of evaluating the impact of short-term combined application of organic and chemical fertilizer on the properties and productivity of summer corn topsoil in medium-yielding meadow-cinnamon soil. Overall, the article is done at a good level. Nevertheless, I believe that the basic SI units, i.e. m or mm, not cm, should be consistently stated throughout the text. Latin strains of bacteria should be labeled in italics. I consider it necessary to supplement the comments in subsection 3.7, which refer to Fig. 9, including the related part of the discussion. The conclusion should be formulated succinctly with an indication of the further use of the results and their possible transfer into cultivation practice. I also recommend doing a language correction. I also recommend revising the cited literature (eliminating a number of old titles) and presenting it in accordance with the standard for bibliometric citations. After incorporating these comments, the text can be accepted for publication.
Author Response
Responses to Reviewer 2’ Comments
The contribution is thematically focused on the issue of evaluating the impact of short-term combined application of organic and chemical fertilizer on the properties and productivity of summer corn topsoil in medium-yielding meadow-cinnamon soil. Overall, the article is done at a good level. Nevertheless,
Comemnt 1: I believe that the basic SI units, i.e. m or mm, not cm, should be consistently stated throughout the text.
Answer: Thank for your comment. However, we don’t quite agree with this comment. Although cm is not the basic SI unit, most of the international journals, including Agronmyo-basel, Agriculture Ecosystems and Environment, Journal of Environmental Management, Soil and Tillage Research, Field Crops Research, Agricultural Water management et al., the units of m, mm cm were all used in its published articles, Therefore, for some units, it is not required to use international uniform units in most international Journal. Please refer to the following references, which the unit of cm is concluded in its papers.
References
Nasi et al., 2023. Can Basic Soil Quality Indicators and Topography Explain the Spatial Variability in Agricultural Fields Observed from Drone Orthomosaics? Agronomy, 13, 669
Whittaker et al., 2023. Nasi et al., 2023. The effects of forage grasses and legumes on subsequent potato yield, nitrogen cycling, and soil properties. Field Crops Research, 290,108747.
Zhao et al., 2021.Effects of different tillage and fertilization management practices on soil
organic carbon and aggregates under the rice–wheat rotation system. Soil and Tillage Research,212, 105071
Ren, et al., 2021. Rhizosphere soil properties, microbial community, and enzyme activities: Short-term responses to partial substitution of chemical fertilizer with organic manure。 Journal of Environmental Management, 299,113650.
Zhang et al., 2021. Responses of maize yield, nitrogen and phosphorus runoff losses and soil properties to biochar and organic fertilizer application in a light-loamy fluvo-aquic soil. Agriculture Ecosystems and Environment, 314,107433.
Comment 2: Latin strains of bacteria should be labeled in italics.
Answer: We agree with this comment. We have checked it throughout the whole manuscript, and have made the specific change in the revised manuscript (Line 26, 276, 279, 408-417, 467 in the revised manuscript).
Comment 3: I consider it necessary to supplement the comments in subsection 3.7, which refer to Fig. 9, including the related part of the discussion. The conclusion should be formulated succinctly with an indication of the further use of the results and their possible transfer into cultivation practice.
Answer: Thank you for your suggestions. According to your suggestions, we have made a further supplementary comment on the subsection 3.7. in the revised manuscript (Line 313-317 in the revised manuscript).
In the discussion section, we have made a discussion on the application of the optimized cultivation practice in the future agriculture production.(Line 456-462 in the revised manuscript)
Comment 4: I also recommend doing a language correction.
Answer: Thank you for your suggestion. I myself and a person proficient in professional English have revised the English grammar of this manuscript
Comment 5: I also recommend revising the cited literature (eliminating a number of old titles) and presenting it in accordance with the standard for bibliometric citations.
Answer: Thank you for your comment. We have revised all the references listed in this manuscript according to the format requirement of Agronomy- basel. (References section in the revised manuscript)
Responses to Reviewer 2’ Comments
The contribution is thematically focused on the issue of evaluating the impact of short-term combined application of organic and chemical fertilizer on the properties and productivity of summer corn topsoil in medium-yielding meadow-cinnamon soil. Overall, the article is done at a good level. Nevertheless,
Comemnt 1: I believe that the basic SI units, i.e. m or mm, not cm, should be consistently stated throughout the text.
Answer: Thank for your comment. However, we don’t quite agree with this comment. Although cm is not the basic SI unit, most of the international journals, including Agronmyo-basel, Agriculture Ecosystems and Environment, Journal of Environmental Management, Soil and Tillage Research, Field Crops Research, Agricultural Water management et al., the units of m, mm cm were all used in its published articles, Therefore, for some units, it is not required to use international uniform units in most international Journal. Please refer to the following references, which the unit of cm is concluded in its papers.
References
Nasi et al., 2023. Can Basic Soil Quality Indicators and Topography Explain the Spatial Variability in Agricultural Fields Observed from Drone Orthomosaics? Agronomy, 13, 669
Whittaker et al., 2023. Nasi et al., 2023. The effects of forage grasses and legumes on subsequent potato yield, nitrogen cycling, and soil properties. Field Crops Research, 290,108747.
Zhao et al., 2021.Effects of different tillage and fertilization management practices on soil
organic carbon and aggregates under the rice–wheat rotation system. Soil and Tillage Research,212, 105071
Ren, et al., 2021. Rhizosphere soil properties, microbial community, and enzyme activities: Short-term responses to partial substitution of chemical fertilizer with organic manure。 Journal of Environmental Management, 299,113650.
Zhang et al., 2021. Responses of maize yield, nitrogen and phosphorus runoff losses and soil properties to biochar and organic fertilizer application in a light-loamy fluvo-aquic soil. Agriculture Ecosystems and Environment, 314,107433.
Comment 2: Latin strains of bacteria should be labeled in italics.
Answer: We agree with this comment. We have checked it throughout the whole manuscript, and have made the specific change in the revised manuscript (Line 26, 276, 279, 408-417, 467 in the revised manuscript).
Comment 3: I consider it necessary to supplement the comments in subsection 3.7, which refer to Fig. 9, including the related part of the discussion. The conclusion should be formulated succinctly with an indication of the further use of the results and their possible transfer into cultivation practice.
Answer: Thank you for your suggestions. According to your suggestions, we have made a further supplementary comment on the subsection 3.7. in the revised manuscript (Line 313-317 in the revised manuscript).
In the discussion section, we have made a discussion on the application of the optimized cultivation practice in the future agriculture production.(Line 456-462 in the revised manuscript)
Comment 4: I also recommend doing a language correction.
Answer: Thank you for your suggestion. I myself and a person proficient in professional English have revised the English grammar of this manuscript
Comment 5: I also recommend revising the cited literature (eliminating a number of old titles) and presenting it in accordance with the standard for bibliometric citations.
Answer: Thank you for your comment. We have revised all the references listed in this manuscript according to the format requirement of Agronomy- basel. (References section in the revised manuscript)
